# Onychomycosis: Old and New

**DOI:** 10.3390/jof9050559

**Published:** 2023-05-12

**Authors:** Narges Maskan Bermudez, Giselle Rodríguez-Tamez, Sofia Perez, Antonella Tosti

**Affiliations:** 1Dr. Philip Frost Department of Dermatology and Cutaneous Surgery, University of Miami Miller School of Medicine, Miami, FL 33125, USA; nmaskan@med.miami.edu (N.M.B.);; 2Dermatology Department, University Hospital “Dr. José Eleuterio González”, Universidad Autónoma de Nuevo León, Monterrey 64460, Mexico

**Keywords:** onychomycosis, dermatophyte, dermoscopy, fungal culture, fungal nail infection, nondermatophyte, yeast

## Abstract

Onychomycosis is a common chronic fungal infection of the nail that causes discoloration and/or thickening of the nail plate. Oral agents are generally preferred, except in the case of mild toenail infection limited to the distal nail plate. Terbinafine and itraconazole are the only approved oral therapies, and fluconazole is commonly utilized off-label. Cure rates with these therapies are limited, and resistance to terbinafine is starting to develop worldwide. In this review, we aim to review current oral treatment options for onychomycosis, as well as novel oral drugs that may have promising results in the treatment of onychomycosis.

## 1. Introduction 

Onychomycosis is a chronic fungal infection of the nail that results in discoloration, onycholysis, and nail plate thickening. The infection most commonly occurs in the toenails and can involve any component of the nail unit, including the nail bed, nail matrix, and nail plate. Onychomycosis affects patients of all ages. However, several studies have established higher prevalence with older age [1,2,3,4]. Other risk factors include diabetes, tinea pedis, poor circulation, immunosuppression, psoriasis, Down syndrome, occlusive footwear, and obesity [5,6,7,8,9].

The worldwide prevalence of onychomycosis is estimated at 10% and accounts for up to 50% of nail diseases [10,11]. Dermatophytes are a common culprit of onychomycosis, with the species *Trichophyton rubrum* and *Trichophyton mentagrophytes* responsible for 60–70% of infections [12]. Yeasts are responsible for approximately 20% of onychomycosis, and non-dermatophytes account for the remaining 10% [13,14]. Studies have demonstrated that mixed infections, non-dermatophytes, and yeasts are more prevalent than previously thought, especially in warmer climates.

Onychomycosis is challenging to treat and is associated with high recurrence rates and treatment failure. Given the limited cure rates with topical antifungals, oral antifungals may be needed in most cases. Oral treatments require lengthy duration of treatment, which poses a risk of adverse effects and drug interactions. Relapse rate can be as high as 25%, and recurrence rates can vary from 6.5% to 53% [15]. Terbinafine and itraconazole are the only approved oral therapies, but fluconazole is commonly utilized off-label. However, complete cure rates (both mycologic clearance and visually clear nails) are limited, and they can range from 35%–55% for terbinafine [16,17,18], 14–43% for itraconazole [16,17,18,19,20], and 21–48% for fluconazole [21,22]. Recently, there are reports of concerning reports of terbinafine resistance in superficial mycoses in India and Europe, and novel agents may play an important role to achieve a cure [23,24,25]. In this review, we aim to describe current treatment options, as well as novel drugs, that may have promising outcomes in onychomycosis.

## 2. Methods

The literature was identified by performing Medline, SCOPUS, Cochrane library, and Google Scholar database searches.

## 3. Clinical Presentation

Patients typically present with white-yellow nail discoloration, hyperkeratosis, onycholysis, and subungual debris. The toenails are the most frequently affected, predominantly the great toenail, with rare involvement of the fingernails [26]. Concomitant tinea pedis or plantar hyperhidrosis are usually found [27,28].

Efforts have been made to establish a clinical classification of onychomycosis. Since the first classification by Zaias [29], additional updates have been proposed [30,31,32]. The clinical classification by Hay and Baran is commonly used among clinicians due to its description of a wide range of subtypes of nail invasion (Table 1) [30].

## 4. Subtypes of Onychomycosis

Distal lateral subungual onychomycosis (DLSO) is the most frequent subtype of onychomycosis and occurs when fungal invasion originates from the distal or lateral undersurface of the nail plate. Clinical features include hyperkeratosis, onycholysis, and white-yellow nail discoloration [26]. However, other color changes, such as black, orange, and brown, have been reported. Additionally, dermatophytomas, which present as white, yellow, orange, or brown longitudinal streaks or patches, may also be seen as complications of dermatophyte infection [33,34].

Superficial onychomycosis (SO) is due to the fungal invasion of the upper surface of the nail plate and typically affects toenails. Clinically, it is characterized either by superficial white or black patches or transverse striae that can be scraped off. The most common etiologic agents are *T. rubrum* [35] and *T. mentagrophytes* [36]. It is often associated with interdigital tinea pedis.

Proximal subungual onychomycosis (PSO) is the result of the invasion of the inner nail plate from fungi penetrating from the proximal nail fold, predominantly by *T. rubrum*. Other implicated organisms include *Candida* [37], *Aspergillus* [38], and *Fusarium* [39]. Clinically, it presents as a white patch of the proximal nail plate, starting from the proximal nail fold or multiple transverse white bands. The superficial nail plate is normal. When paronychia is associated, *Candida* species or molds are commonly implicated [40]. This subtype of onychomycosis is also common among immunosuppressed individuals.

Endonyx onychomycosis (EO) is the consequence of direct invasion of the distal nail plate. It presents as a lamellar nail splitting with milky patches. The main etiologic agents are *T. soudanense* and *T. violaceum* [30,31].

Mixed pattern onychomycosis (MPO) occurs when more than one pattern of nail plate fungal infection is found within the same nail. The most common combination is DLSO with SO, or PSO with SO. A wide range of etiologic agents may be implicated (Table 1) [30].

Total dystrophic onychomycosis (TDO) is the end stage of chronic onychomycosis, mainly DLSO or PSO. Clinically, it is characterized by nail plate crumbling and thickening of the nail bed with debris [30]. The most common agents are dermatophytes (*T. rubrum*), as well as some molds [30].

## 5. Secondary Onychomycosis

This is a clinical scenario when fungal invasion occurs secondary to a non-infectious nail condition, such as trauma, psoriasis, lichen planus, etc. Clinical features of the underlying nail condition are associated with hyperkeratosis, nail discoloration, onycholysis, patches, and striae. This subtype is a common presentation, and diagnosis may be challenging.

## 6. Diagnosis

Clinical signs of nail thickening, yellow, white, or brown nail discoloration, onycholysis, subungual debris, and multiple nail involvement, should prompt clinicians to assess for onychomycosis. Dermoscopy findings in onychomycosis include jagged proximal edges with spikes, longitudinal striae, a ruined appearance, and longitudinal ridges along the nail bed [41]. Dermoscopy can also be used to assess for trauma or other nail disorders, such as nail psoriasis, subungual warts, and lichen planus.

Diagnosis by clinical features is not always reliable, and laboratory isolation of fungi continues to be the diagnostic gold standard. A sensitive test for onychomycosis includes nail clippings sent in formalin for histopathology and periodic acid-Schiff (PAS) stains and other fungal stains. Alternatively, potassium hydroxide (KOH) preparation of nail scrapings can also be performed to visualize hyphae. While both the PAS stain and KOH preparation are sensitive tests, neither method can identify specific pathogens. Fungal culture can be used for identification of pathogens; however, its sensitivity is low, and there is a risk of false-positive results due to contamination.

Polymerase chain reaction (PCR) is an excellent diagnostic method that has high sensitivity, but false positives are common. Given that the number of mixed onychomycosis and non-dermatophyte infections are becoming more prevalent, molecular diagnosis can aid in the selection of appropriate antifungal treatment.

The presentation of onychomycosis can also help determine areas of the nails needed for the sample collection. Nail clippings should be at least 4 mm to maximize diagnostic accuracy [42]. For DLSO, samples should be taken from the nail bed in the proximal area of involvement where the concentration of hyphae will be the highest [43]. In white superficial onychomycosis (WSO), a specimen should be collected by scraping the superficial aspect of the nail with a No. 15 scalpel [8]. In PSO, the proximal nail can be obtained with a 3 mm punch of the proximal nail plate [43,44].

## 7. Defining a Cure

Onychomycosis studies often report the mycologic cure, clinical cure, and complete cure rates. A mycologic cure is achieved when both the culture and direct microscopy are negative after medical treatment. A clinical cure is defined as a normal appearance of the affected nail. A complete cure is defined as both negative mycology and absence of clinical signs in the nail. The goal of treatment is complete cure; however, patients often have nail abnormalities before the development of the fungal infections and will not achieve fully normal nails after treatment. In this review, we will focus primarily on complete cure rates.

## 8. Treatment Approach

When approaching onychomycosis treatment, the three main pharmacologic strategies include oral treatment, topical treatment, or combination therapy.

Topical treatment of onychomycosis is complicated by several factors, including limited access to the nail bed and complex pharmacologic properties needed to allow for nail plate penetration. Patients who have nail plate thickening and onycholysis particularly add to the suboptimal nature of topical drug transportation. Nail lacquers represent a unique pharmacologic property that allows for higher levels of diffusion across the nail plate as its drug concentration increases, while the solvent evaporates [45]. Leading nail lacquer products, ciclopirox 8%, amorolfine 5%, and efinaconazole 10%, have reported complete cure rates of 5.5–8.5%, 15.2–17.8%, and 15.2–25.6%, respectively [46,47,48,49,50]. Despite lower cure rates, topical monotherapy is recommended in mild toenail onychomycosis limited to the distal nail, as well as WSO. Recent data, however, show that efinaconazole can be very effective in treating onychomycosis complicated by dermatophytomas and longitudinal spikes [51,52].

While higher complete cure rates are seen in oral antifungal monotherapy than topical monotherapy, topical treatment can play a role in maintenance therapy after the completion of oral antifungal therapy. Oral therapy is generally recommended for patients with DLSO involving more than 50% of the nail, DLSO affecting more than two nails, PSO, and deep WSO [53].

Oral antifungal agents available for the treatment of toe and fingernail onychomycosis include terbinafine, itraconazole, and fluconazole. Terbinafine monotherapy with continuous 250 mg dosing remains the current first-line treatment recommendation. Terbinafine pulse dosing, itraconazole, and fluconazole therapy are second-line treatments that are useful when there are contraindications with terbinafine use.

## 9. Oral Treatments

Several treatment options exist for the treatment of onychomycosis (Table 2). Oral terbinafine is the most effective Food and Drug Administration (FDA)-approved treatment for onychomycosis [54]. Terbinafine treatment duration is typically a minimum of six weeks and twelve weeks for fingernail and toenail onychomycosis, respectively. Terbinafine complete cure rates vary between 35% and 78% in patients with toenail onychomycosis [16,17,18,55]. Terbinafine sensitivity is not well studied in onychomycosis involving non-dermatophyte molds or yeast [56].

Terbinafine pulse dosing regimens can vary in dose, duration, and frequency. A common regimen is two pulse regimens of terbinafine 250 mg daily for four weeks, followed by four weeks off. A meta-analysis by Gupta et al. determined that continuous terbinafine regimen is generally superior to pulse dosing for mycologic cure. However, both continuous and pulse dosing had similar complete cure rates [57,58]. Other pulse regimens for oral terbinafine include 500 mg daily for one week, followed by three weeks of no treatment, repeated every month for three months [59]. Studies assessing cost [60] and compliance [61] did not identify a difference between continuous and pulse regimens. However, pulse dosing may be preferred due to patient preference, side effects, comorbidities, and risk of potential drug–drug interactions [57,60].

Contraindications to terbinafine use include its interaction with other pharmaceuticals that are metabolized by cytochrome P-450 enzyme 2D6. Most notably, metoprolol is among this category of cP450 2D6 metabolized drugs, making terbinafine treatment discouraged in patients taking metoprolol. In these cases, oral treatment options for onychomycosis include itraconazole, an alternative broad-spectrum antifungal. Itraconazole is FDA-approved for the treatment of onychomycosis and is effective against dermatophytes, yeasts, and non-dermatophyte molds. Dosing options include a continuous treatment regimen of 200 mg daily for three months or a four-pulse treatment regimen of 400 mg daily for one week, followed by a three-week pause in drug treatment. Complete cure rate with itraconazole treatment ranges between 14–43% [16,17,18,19,20].

An additional second-line treatment option includes the oral antifungal fluconazole. Fluconazole is not FDA approved for the treatment of onychomycosis. However, clinical trials have established its efficacy in dermatophyte nail infections. Typical dosing regimen of fluconazole is 150–450 mg once weekly for a duration of six months for fingernails and 12 months or more for toenails [22,62]. Complete cure rates range from 21% to 48%, depending on dosage and treatment duration [21,22].

**Table 2 jof-09-00559-t002:** Current oral antifungal therapies.

Oral Antifungal	Dosing Regimen	Complete Cure Rate (%)
Terbinafine [16,17,18,55]	Fingernail: 250 mg/d for six weeks Toenail: 250 mg/d for 12–16 weeks Pulse: 250 mg/d for four weeks on, four weeks off, four weeks on	35–78
Itraconazole [16,17,18,19,20]	Continuous: 200 mg/d for 12 weeks Pulse: 400 mg/d for one week per month for 16 weeks	14–43
Fluconazole [21,22,62]	Fingernail: 150–450 mg/week for six months Toenail: 150–450 mg/week for 12 months	21–48

Antifungal resistance is a variable that also should be considered when choosing treatment options. As dermatophytic infections evolve over time, the efficacy of oral antifungals may change, as well. In a recent clinical trial, commonly prescribed antifungals, including terbinafine, itraconazole, and fluconazole, were tested against chronic and chronic relapsing tinea corporis, cruris, and faciei [25]. After four weeks of treatment, all drugs demonstrated a cure rate around 8% or less. After eight weeks of treatment, cure rates with terbinafine, itraconazole, and fluconazole were reported as 28%, 66%, and 42%, respectively. This study indicates the reality of antifungal resistance, as well as evidence suggesting the effectiveness of itraconazole in terbinafine-resistant dermatophytosis. Similar reports of increasing terbinafine resistance in *Trichophyton species* have been reported in Europe [23,24]. More studies are needed to further assess treatment regimens in multi-drug resistant cases.

Due to the systemic nature of oral antifungal medications, side effects and drug–drug interactions should be considered when prescribing each of these medications. Terbinafine has been associated with hepatic injury. Therefore, liver function tests are recommended prior to starting treatment, although some doctors recommend evaluating liver function throughout the course of terbinafine treatment [20]. Itraconazole therapy carries a high risk of drug interactions and should be used cautiously in patients with cardiac conditions due to its risk of heart failure and arrythmias [63]. Fluconazole is also associated with high risk of interactions (not much when used as pulse treatment) cardiovascular risk and has been reported to prolong the QT interval [64]. Additional side effects with fluconazole include significant congenital defects, and avoidance of this medication during pregnancy is strongly advised [65].

Interval monitoring in oral antifungal treatment is unnecessary in adults and children without preexisting hematologic or hepatic abnormalities [66]. The recommendation for continued hepatic function monitoring while receiving treatment with an oral antifungal has since been removed by the FDA. Due to higher incidence rates of drug-induced liver injury seen in patients with preexisting hepatic conditions, dosing adjustments and continued interval monitoring is beneficial in these cases.

Poor response to treatment is seen at higher rates in individuals with subungual keratosis measuring more than 2 mm and fungal infection involving the lateral nail or over 50% of the entire nail unit [15]. Recently a few studies showed that topical treatment with efinaconazole can be useful in this setting [51,52]. Given the terbinafine sensitivity to dermatophytes, onychomycosis caused by mixed infection or resistant organisms are also associated with poor prognostic outcomes. In the setting of decreased peripheral circulation, as in elderly or diabetic patients, poor treatment response to first-line oral antifungals has been reported.

## 10. Special Populations

Onychomycosis is more common in adults than in children. Prevalence-based studies are conflicting, with some reports ranging from 0.2% to as high as 7.66% [67,68]. While previously considered rare, several studies suggest that onychomycosis in children is on the rise [69,70,71,72]. While there are no FDA-approved medications for pediatric patients, evidence suggests that current treatments are safe in children, and dosing should be readjusted according to weight.

Onychomycosis has a higher incidence among elderly patients, with a reported prevalence nearly 20% in patients greater than 60 years old [73]. Few studies have evaluated the efficacy of oral antifungals in the elderly population [74,75,76]. Cure rates with approved therapies terbinafine and itraconazole have been reported to be lower when compared to adult population [16,76,77]; however, they are considered a safe option when possible, and drug interactions have been evaluated [75].

Immunocompromised patients, such as HIV positive individuals with CD4 counts less than 500, have a higher prevalence of tinea pedis and onychomycosis [78,79]. There are scarce studies that have evaluated oral antifungals in the HIV population [74,75]; however, terbinafine has shown to be efficacious and well tolerated [75]. Of note, HIV positive individuals who are on antiretroviral therapy with normal CD4 counts can eradicate the infection. Additionally, combined antiretroviral therapy itself has demonstrated to clinically improve onychomycosis [80].

## 11. Terbinafine Resistance: A Public Health Concern

Terbinafine resistance is an emerging problem globally, and isolates have been documented in India and Europe with increasing frequency [23,24]. The Trichophyton species is commonly identified, specifically *T. rubrum*, *T. interdigitale*/*mentagrophytes*, and *T. indotineae* [23,81,82,83]. Terbinafine inhibits squalene oxidase and interferes with ergosterol production, a compound necessary for fungal plasma membrane structure. Trichophyton resistant cases arise when point mutations develop in the squalene oxidase gene [81,82,83,84,85]. While cases of antifungal resistance have been largely reported in dermatophytic infections of the skin, the emergence and spread of these organisms is an important public health concern that can have significant consequences in onychomycosis cases.

As with antimicrobial resistance, dermatophyte-resistant species can be secondary to natural microbial changes over time, increased exposure to antifungals, and/or noncompliance. Mycological identification in suspected onychomycosis cases is crucial to avoid unnecessary antifungal and steroid use, and compliance should be emphasized in patients who begin onychomycosis therapy. In cases of recalcitrant infections, antifungal susceptibility testing (AFST) should be explored to guide management. Additionally, azole-based treatments with itraconazole, longer treatment duration, and/or combination therapies may be necessary to eradicate terbinafine-resistant infections.

## 12. Novel Onychomycosis Therapies

New oral antifungal agents show promising results in the treatment of onychomycosis. Many of these are azole-based treatments. However, these medications are not approved by the FDA for the treatment of onychomycosis, and some are not available in the United States. As challenges continue to rise with terbinafine resistance, these new therapies may be promising in the treatment of superficial mycoses, such as onychomycosis (Table 3). 

Fosravuconazole L-lysine ethanolate is a ravuconazole prodrug approved in Japan for the use of onychomycosis. A phase III trial demonstrated a complete cure rate of 59.4% with fosravuconazole 100 mg daily for 12 weeks [86]. Fosravuconazole also has a preferable safety profile in the elderly due to less potent inhibition of cytochrome P450 [87].

Posaconazole is a broad-spectrum azole that is FDA-approved for invasive *Aspergillus* and *Candida* infections and oral pharyngeal candidiasis. A phase IIB trial by Elewski et al. evaluated the use of posaconazole in onychomycosis and demonstrated a complete cure rate of 51.4% with posaconazole 200 mg daily for 24 weeks [88]. The efficacy and safety profile are favorable. However, the cost of posaconazole and lack of FDA approval for onychomycosis limits its use.

Oteseconazole, also known as VT-1161, is a tetrazole antifungal that is FDA-approved for recurrent vulvovaginal candidiasis. A phase II trial assessed oteseconazole for distal and lateral subungual dermatophyte onychomycosis and reported mycologic cure rates between 41–45% at 60 weeks. Patients received either oteseconazole 300 mg once daily for two weeks, followed by 300 mg once weekly for 10 or 22 weeks, or 600 mg once daily for two weeks, followed by 600 mg once weekly for 10 or 22 weeks [89].

Voriconazole is a broad spectrum, triazole antifungal FDA approved for invasive *Aspergillosis*, candidemia in non-neutropenics, other deep tissue *Candida* infections, esophageal candidiasis, and Scedosporiosis and Fusariosis. A prospective clinical trial assessed the use of voriconazole for onychomycosis and reported a complete cure rate of 67.9% with eight weeks of therapy (200 mg twice daily for four weeks, followed by 200 mg once daily for four weeks) [90]. Dose adjustments may be needed in patients with hepatic or renal impairment [91].

Albaconazole is a new broad-spectrum antifungal with activity against dermatophytes and yeasts. Its efficacy in onychomycosis has been described in randomized clinical trials, as well as recently in a systematic review. Doses range from 100 to 400 mg once weekly for a 24–36 weeks treatment period. Its efficacy is dose-dependent, with the highest complete cure rate at 400 mg once weekly for 36 weeks (33%). However, further studies are needed to evaluate its security profile [92,93]. Albaconazole is not currently FDA-approved for any indication.

## 13. Other Treatments

Oral therapies for the treatment of onychomycosis can be augmented with the use of adjunct treatments, such as lasers, photodynamic therapy (PDT), or nail trimming/debridement. Antifungal combination therapy has also shown efficacy in treatment advancement.

Several studies have analyzed the use of the long pulsed neodymium-doped yttrium aluminum garnet (Nd:YAG) laser, diode laser, and fractional carbon dioxide (CO_2_) laser in onychomycosis. Lasers are currently FDA-approved for mild temporary clearance of the nail. It is theorized that lasers can be fungicidal through photothermolysis, with rapid elevation of temperature leading to fungal cell death. However, randomized trials yielded poor results with no statistical difference in patients who had laser therapy compared to placebo [94,95]. A study by Lim et al. did show improvement in onychomycosis when lasers were used as combination therapy with topical amorolfine for 12 weeks. The authors concluded that the beneficial effects may be through a combination of direct fungicidal effects and nail changes by the lasers, allowing deeper penetration of the topical drug [96]. Therefore, lasers can be considered as an adjunct therapy in elderly patients, patients with hepatic or renal disease, or other contraindications, but patients should be cautioned of the limited clinical improvement and high costs associated with lasers [97].

PDT is a non-invasive therapy that uses photosensitizing agents and a light source to generate reactive oxygen species and subsequently destroy cells in a given area. Many dermatophytes responsible for onychomycosis can absorb photosensitizing agents, making them susceptible to apoptosis by PDT [98]. PDT itself presents an optimal treatment option for patients with contraindications to oral antifungals. PDT used in combination with oral antifungals has also shown higher cure rates and a shorter treatment duration [99]. The most common photosensitizing agents used in PDT for onychomycosis are aminolevulinic acid (ALA), methyl-aminolevulinic acid (MAL), and methylene blue (MB). The number of treatment sessions varies from three to twelve sessions with incubation times of 1–5 h. The efficacy of MAL and MB use in PDT in conjunction with oral terbinafine demonstrated a 70% complete cure rate in both modalities [100].

As an additional treatment option, nail debridement can be used in combination with oral antifungals. Patients who underwent aggressive nail debridement with oral terbinafine therapy demonstrated higher clinical cure rates (59.8% vs. 51.4%) and complete cure rates than patients who received terbinafine therapy alone [101]. Data published on this treatment regimen, however, are relatively limited.

Combination therapy of antifungals has been shown to improve treatment response compared to monotherapy. In addition to improved efficacy, combination therapy may also help to combat antifungal resistance, which is being encountered at an increasing frequency [102]. However, more studies are needed to assess oral antifungal combination therapy [103].

## 14. Conclusions

Onychomycosis is a common nail disease caused by dermatophytes, yeasts, and NDMs. Oral treatment is indicated in moderate to severe cases, multiple digit involvement, and/or failure of topical therapies. Although several antifungal oral therapies are available, terbinafine-resistant isolates are emerging and can have significant impact on future management. Novel drugs are needed to combat this challenge, and several studies have demonstrated promising preliminary data with newer antifungal therapies. Additional studies are needed to assess and analyze safety profiles, dosages, and establish guidelines for these new drugs.

## Figures and Tables

**Table 1 jof-09-00559-t001:** Onychomycosis subtypes and their associated clinical features and etiologic agents.

Onychomycosis Subtypes	Clinical Features	Etiologic Agents
Distal and lateral subungual onychomycosis (DLSO) [26,33,34]	○ Hyperkeratosis ○ Onycholysis ○ Nail discoloration (white-yellow, black, brown, or orange)	Dermatophytes (*T. rubrum*, *T. mentagrophytes*, *E. flocossum*) *Fussarium species* *Candida albicans* *Scopulariopsis brevicaulis*
Superficial onychomycosis (SO) [35,36]	○ Superficial patches (white or black) ○ Transverse striae	Dermatophytes (*T. rubrum*, *T. metagrophytes*) *Fussarium species* *Scytalidium*
Proximal subungual onychomycosis (PSO) [37,38,39,40]	○ Transverse white bands ○ Proximal white patch ○ Superficial nail is normal. ○ Paronychia (as a preceding condition)	*T. rubrum* *Fusarium species* *Candida albicans* *Aspergillus species*
Endonyx onychomycosis [30,31]	○ Lamellar nail splitting ○ Nail discoloration (milky patches)	*T. soudanense* *T. violaceum*
Mixed pattern onychomycosis [30]	○ DLSO + SO features (hyperkeratosis, onycholysis, nail discoloration, superficial patches or transverse striae) ○ PSO + SO features (transverse bands, longitudinal bands, transverse striae and superficial patches)	*T. rubrum* *Fussarium species*
Total dystrophic onychomycosis (TDO) [30]	○ Dystrophic and crumbled nail plate ○ Nail bed thickening	Dermatophytes (*T. rubrum)* *C. albicans*
Secondary onychomycosis	○ Clinical features of underlying nail condition + onychomycosis findings (hyperkeratosis, discoloration, patches, striae)	Dermatophytes Molds

**Table 3 jof-09-00559-t003:** Novel antifungal therapies and their recommended dosage for the treatment of onychomycosis.

Oral Antifungal	Dosing Regimen *	Complete Cure Rate (%)	Notes
Fosravuconazole [86,87]	100 mg/d for 12 weeks	59.4	Safe in elderly due to less potent inhibition of CP450 Approved in Japan only. No clinical data on children, pregnant women, or nursing mothers.
Posaconazole [88]	200 mg/d for 24 weeks	51.4	Should be avoided in pregnancy. Multiple drug interactions
Oteseconazole [89]	Loading dose: 300 mg/d for two weeks. Followed by 300 mg/week for 10 or 22 weeks OR Loading dose: 600 mg once daily for two weeks. Followed by 600 mg/week for 10 or 22 weeks	41–45	No clinical data on children, pregnant patients, or nursing mothers.
Voriconazole [90,91]	200 mg/bid for four weeks, followed by 200 mg/d for four weeks	67.9	Dose adjustments needed in patients with hepatic or renal impairment. Multiple drug interactions
Albaconazole [92,93]	400 mg/week for 36 weeks	33	Not available in the USA Lengthy therapy regimen

* All dosing regimens are for toenail onychomycosis.

## Data Availability

Not applicable.

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
