# Peer review of "Onychomycosis: Old and New"

_jof, 2023, doi:10.3390/jof9050559_

Round 1
Reviewer 1 Report
Your manuscript is a comprehensive review of onychomycosis. The descriptions of the clinical subtypes of onychomycosis, the organisms involved, and laboratory diagnostic strategies are excellent. The discussion of potential future treatments currently being studied for onychomycosis is enlightening.
The formatting of Table 1 is challenging to follow due to the abnormal alignment of the text, and it would be easier to read if the columns were justified correctly.
Author Response
Thank you for your feedback. The tables and columns have been restructured and organized according to your comments.
Reviewer 2 Report
Article Type: Review Article
O objetivo do manuscrito foi avaliar os tratamentos orais da onicomicose. Na minha opinião o manuscrito precisa de algumas alterações para publicação.
1) Add references to tables.
2) The focus of the work is oral treatment, however, the authors also addressed topical treatment and other therapeutic options. I suggest changing the title or adjusting the manuscript, as the authors addressed onychomycosis very comprehensively, which is commonly observed in other publications.
3) I suggest that the author addresses more about the new therapeutic options, topic: “Novel Onychomycosis Therapies”, as it is the differential of the work.
4) Add the methodology. What were the criteria used to select the articles?
Author Response
Thank you for taking the time to review our manuscript. The following changes have been made based off your feedback.
- References have been updated on the tables.
- The title has been changed to "Onychomycosis: Old and New"
- See above
- A methods section has been added
Thank you for your time!
Reviewer 3 Report
Excellent, thorough, and timely review--I am glad you added in a paragraph regarding terbinafine resistance and that you reviewed the newer oral antifungal azoles.
Author Response
Thank you for your feedback, we appreciate it!
Round 2
Reviewer 2 Report
Article accepted for publication.